
# Spatial and temporal subsidence characteristics in Wuhan city (China) during 2015-2019 inferred from Sentinel-1 SAR Interferometry

Xuguo Shi[1], Shaocheng Zhang[1], Mi Jiang[2], Yuanyuan Pei[3], Tengteng Qu[4,5,6], Jinhu Xu[1], Chen Yang[7]

[1] School of Geography and Information Engineering, China University of Geosciences, Wuhan, 430074, China
[2] School of Geospatial Engineering and Science, Sun Yat-Sen University, Guangzhou, 510275, China
[3] School of Civil Engineering, Anhui Jianzhu University, Hefei, 230601, China
[4] College of Engineering, Peking University, Beijing, 100871, China
[5] China-Pakistan Joint Research Center on Earth Sciences, Islamabad, 45320, Pakistan
[6] State Key Laboratory of Geohazard Prevention and Geoenvironment Protection, Chengdu University of Technology, Chengdu 610059, China
[7] Institute of karst geology, CAGS/ Key Laboratory of Karst Dynamics, MNR & GZAR, Guilin, 541004, China

*Correspondence to*: Xuguo Shi (shixg@cug.edu.com)

**Abstract.** Ground subsidence is regarded as one of the most common geohazards accompanied with the rapid urban expansion in recent years. In the last two decades, Wuhan located in the alluvial Jinaghan Plain has experienced great urban expansion with increased subsidence issues, i.e. soft foundation subsidence and karst collapses. Here we investigated subsidence rates in Wuhan city with 2015-2019 Sentinel-1 SAR images. We found that the overall subsidence over Wuhan region is significantly correlated with the distribution of engineering geological regions (EGSs). We further validated the InSAR measurements with better than 5 mm accuracy by comparing with levelling measurements. Subsidence centres in Qingling-jiangdi, Houhu, Qingshan and Dongxihu area were identified with displacement rates of approximately 30 mm/yr. Our results demonstrated that the dominant driven factor is ongoing constructions and the subsidence centres shifted with construction intensities. Qingling-Jiangdi area in our study is a well-known sites of karst collapses. We find the nonlinear subsidence of this area is correlated with the water level variations of the Yangtze River.

## 1 Introduction

Nowadays, land subsidence have become a serious problem along with the rapid urban expansion (Xue et al. 2005). Land subsidence events have been reported in major cities all over the world (e.g. Shanghai (Perissin et al. 2012), Beijing (Hu et al. 2019, Zhou et al. 2019), Seville (Ruiz-Constán et al. 2017), Texas (Kim et al. 2019), Hanoi (Dang et al., 2014) and Jakarta (Ng et al. 2012, Chaussard et al. 2013)). Over 50 cities in China have been suffering from land subsidence due to various factors (Yin et al. 2005). The major causing factors of urban land subsidence are extensive pumping of groundwater (Xue et al. 2005, Yin et al. 2005), ground fissures (Zhao et al. 2018) and tectonic faults (Xue et al. 2005, Hu et al. 2019) which threat the normal operation of urban systems and people's daily lives. The approximately accumulated economic losses caused by





land subsidence reached up to 450-500 billion RMB during 1949 ~ 2005 in China (Yin et al. 2005). Therefore, great efforts need to be made to monitor and reduce the land subsidence and related issues.

SAR Interferometry (InSAR) provides a unique tool for the quantitative measurement of the Earth's surface deformation with
wide coverage and high resolution (B ürgmann et al. 2000). The phase information recorded by complex SAR images can be used to measure subtle displacements with millimetre to centimetre accuracy level. However, the temporal and geometric decorrelations, the DEM uncertainties and atmospheric turbulence effect make it difficult to achieve such high accuracy. These challenges was overcome along with proposed time series InSAR analysis methods (e.g. the Persistent scatterers InSAR (PSI) (Ferretti et al. 2001), the small baselines subset method (SBAS) (Berardino et al. 2002), the Quasi persistent
scatterers InSAR (Perissin and Teng 2011), the SqueeSAR$^{TM}$ (Ferretti et al. 2011, Jiang and Guarnieri 2020) and Coherent scatterer InSAR (Dong et al. 2018) by analysing the stable or slowly decorrelated pixels in multi-temporal SAR images. The new generation of SAR sensors (e.g. Sentinel-1, ALOS2 PALSAR2, TerraSAR-X, COSMO-Skymed and Radarsat-2) are now regularly operate in orbit with very short revisit time which enable InSAR achieve wide-area monitoring of ground displacement at high precision (Gee et al. 2019).

As the largest city in central China, Wuhan has experienced significant urban expansion in the last two decades (Tan et al. 2014). The ongoing constructions of high rise buildings and metro-lines all over the metropolitan area have induced serious subsidence (Zhou et al. 2017). As a result, the subsidence areas in Wuhan have remarkably extended. Recent years, more than 40 communities in Jiang'an District, Jianghan District, Qiaokou District and Wuchang District in Wuhan City have experienced ground subsidence, resulting in cracks in buildings and municipal roads in the varying degrees (Guan et al.
2016). Although over 300 benchmarks have been set to understand the subsidence trends (Zhou et al. 2017), the coverage of the benchmarks are too sparse to capture the global picture of deformation patterns. Thus, the time-series InSAR techniques making use of the stable pixels in SAR images can make up for this limitation. High resolution TerraSAR-X and COSMO-Skymed SAR images are used to investigate the spatial and temporal subsidence of Wuhan urban area during 2009-2010 (Bai et al. 2016), 2013-2014 (Costantini et al. 2016), 2013-2015 (Bai et al. 2019). They found the subsidence velocity in
Houhu area reached over -70mm/yr and are mainly correlated with construction activities on quaternary soft clay and carbonate rocks areas (Costantini et al. 2016, Bai et al. 2019). Similar results were also obtained by Radarsat-2 SBAS-InSAR analysis from 2015-2018 (Zhang et al. 2019) and medium resolution Sentinel-1 images PSI (Benattou et al. 2018) or SBAS-InSAR (Zhou et al. 2017) analysis from 2015~2017. According to geological investigations (Guan et al. 2016, Li et al. 2019), the subsidence might be correlated with engineering geological zones which are seldom studied.

In this study, 113 Sentinle-1 SAR images from April 2015 to September 2019 covering the Wuhan metropolitan area are analysed with SBAS-InSAR method. Comparison between InSAR and leveling measurements are conducted to validate the reliability of our measurements. We found that spatial subsidence patterns are correlated with distributions of engineering geological zones in the first terrace in Wuhan city. Relationships between time series subsidence and rainfall/ river level are also discussed.


## 2 Study area and datasets

### 1.2 The Wuhan metropolitan area

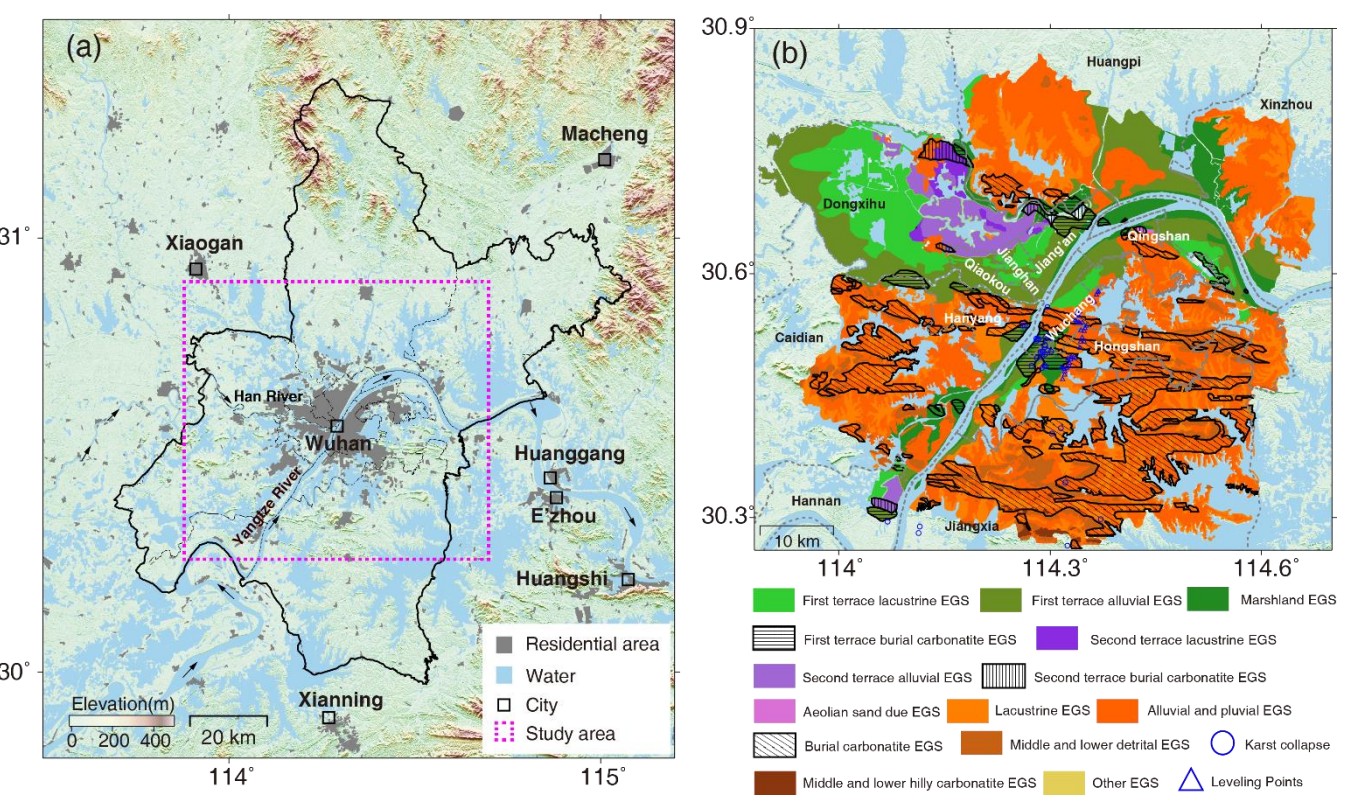

**Figure 1: (a) Topography of our study area, (b) engineering geological subregions (EGSs) over Wuhan metropolitan area and distribution of historical karst collapse. Modified from Li et al., (2019) and Zheng et al., (2019).**

Wuhan, the capital city of Hubei province, plays an important role in industry, science, education and transportation of China. It is located eastern of the alluvial Jianghan Plain formed by Yangtze River and its largest tributary, the Han River. The rivers and scattered lakes make 1/4 of Wuhan's urban area occupying by water as shown in Fig.1 (a). Since the launch of "The Rise of Central China" in 2004, Wuhan has experienced rapid economic growth and urban sprawl. The annual urban expansion velocity reached 46.75%, and the urban areas increased from $4.19 \times 10^4$ ha in 1988 to $49.29 \times 10^4$ ha in 2011 (Tan et al. 2014). Fig. 1(a) shows the built-up area in 2015 derived from 1:250,000 national basic geographic database (http://webmap.cn/main.do?method=index) released by the National Geomatics Centre of China.

Our study area covers the Wuhan metropolitan area as indicated by the dashed rectangle in Fig. 1(a). The terrain is low and flat with maximum elevation of 240 m. Over 95% of our study area are covered by Quaternary layers (Deng et al. 1991, Xu 2016, Li et al. 2019) with diverse lithology, including gravel, sand (coarse sand, fine sand, silt), sub-sand, sub-clay, clay, muck etc. (Table S1). Moreover, the areas of burial or covered dissolution carbonate rocks (a.k.a Karst) are 1091.51 km², 





accounting for 12.85% of the total area of Wuhan (Tu et al. 2019, Zheng et al. 2019) as shown in Fig. 1(b). Thirty-eight karst collapses (a.k.a sinkholes) marked with blue circles in Fig. 1(b) were recorded in Wuhan between 1931-2018 caused by anthropogenic activities and natural forces (Tu et al. 2019). Twenty-seven of the karst collapses are occurred after 2005 and

only two of them are caused by natural factors (Tu et al. 2019). Geological investigations are conducted to understand the engineering conditions for urban construction or subsidence mechanisms (Li et al. 2019, Zheng et al. 2019). They divided the metropolitan area into 4 engineering geological zones (EGZs) and 13 sub-regions by considering the geomorphologic characteristics, Quaternary geological characteristics (e.g. stratum, genesis, lithology, etc.) and engineering geological properties of soils.

The engineering geological map of Wuhan metropolitan area is shown in Fig. 1(b). The 4 EGZs are first terrace EGZ, second terrace EGZ, wavy hillocky EGZ, denuded hilly EGZ, which accounts for approximately 30.27%, 4.39%, 60.09% and 4.71% of the area of Wuhan metropolitan area. The detailed description of 13 engineering geological sub-regions (EGSs) are given in Table S1. We should note that soft soil is a general term of muck and muck soil. It has characteristics of high water content, large void ratio, high compressibility, low shear strength, poor bearing capacity, small consolidation coefficient,

long consolidation time and poor water permeability etc. The thickness of soft soil in the first terrace generally ranges from 1 ~ 18.5 m and the maximum value reaches 37 m while the thicknesses of soft soils are relatively low in the other EGSs ranging from 1 ~ 15 m (Wuhan Bureau of Natural Resources and Planning 2018).

**2.2 Datasets**

The Sentinel-1 satellite constellation conducted by the European Space Agency (ESA) composed of Sentinel-1A launched

on 3 April 2014 and Sentinel-1B launched on 25 April 2016. It is the first sensor utilizing the Interferometric Wide swath (IW) Mode as the main acquisition mode characterized by 5 by 20 meters resolution with swath width of 250 km. 113 ascending track Sentinel-1A IW SAR images between 11th April 2015 and 29th September 2019 were acquired as shown in Fig. 2. The ALOS World 3D 30 m (AW3D30) DSM release by the Japan Aerospace Exploration Agency (JAXA) (Takaku et al. 2016) are used for co-registration, differential interferogram generation and geocoding. Measurements from 38 levelling

points marked with triangles in Fig. 1(b) obtained at 10th September 2016, 10th March 2017, 10th October 2017 and 10th May 2018 are used to validate our InSAR measurements.

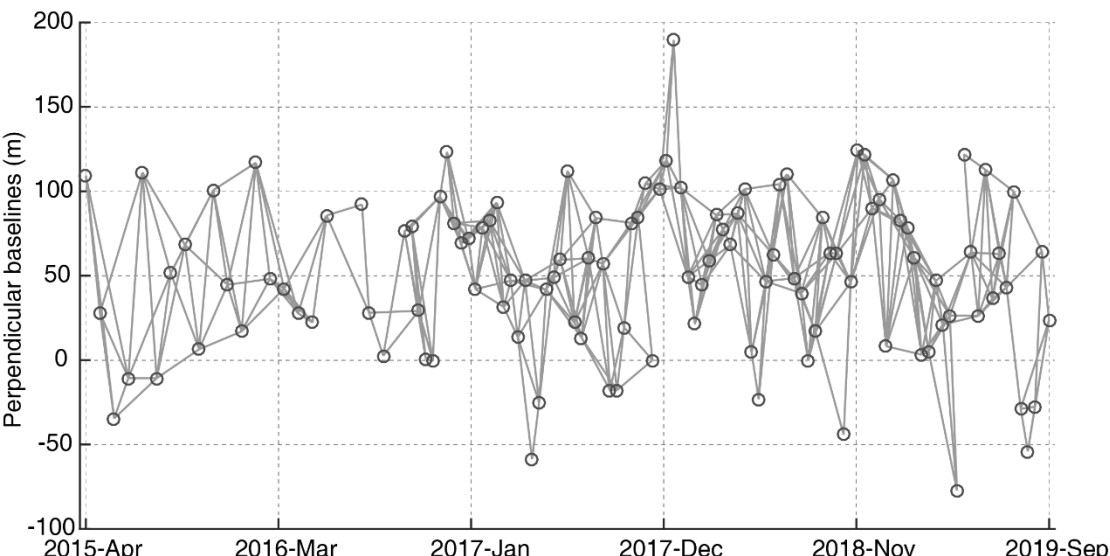

**Figure 2: Graph of the temporal network used for InSAR time-series analysis.**

## 3 Time-series Sentinel-1 SAR interferometry analysis

### 2.1. Interferomeric processing

Assume we have collected a stack of Sentinel-1 IW Single look complex (SLC) images which are generally consist of three subswaths recording subsets of echoes of the SAR aperture (Torres et al. 2012). These echoes are basic units called bursts. There are overlaps between adjacent bursts and sub-swathes for the convenience of synchronizing the bursts to a mosaicked image without gaps (Yague-Martinez et al. 2017). Bursts contained in the subswaths in Geotiff format are extracted with corresponding annotation files. A primary image was selected by considering the distribution of temporal and spatial baselines. In our study, the image obtained at 26th, November 2017 was selected. The following interferomeric processing was performed at burst level.

Due to the significant variations of Doppler centroid frequency in the burst, high level co-registration accuracy with better than 0.001 pixels are required for the interferomeric analysis (Jiang, 2020). In our study, an AW3D DSM and Sentinels Precise Orbit Determination (POD) service were first used for geometric co-registration between consecutive bursts. Then, a network-based enhanced spectral diversity approach was employed to estimate time series azimuth shifts (Jiang, 2020). Burst de-ramping, re-ramping and resample were finally carried out to resample bursts. Individual bursts can be then be merged into seamless SLCs. Images with small temporal (< 60 days) and perpendicular baselines (< 500 m) are combined to generate 368 differential interforgrams as shown in Fig. 2.




## 2.2. Time series displacement retrival

The SBAS-InSAR makes use of point-like targets which remain high level of coherence over a long temporal period or slow decorrelated pixels which will remain coherent in a short period (Hooper 2008, Jiang and Guarnieri 2020). Amplitude dispersion value is used to initially select the candidates that can be used to extract useful signals (Ferretti et al. 2001). Phase

stability analysis was firstly performed on these candidates. The final pixels used for the displacement rate estimation were determined by temporal coherence threshold with 0.3. Then, 3D phase unwrapping were further performed to retrieve continuous phase in the spatial and temporal dimension (Hooper and Zebker 2007). Generally, there are orbital phase ramps, DEM residual and atmospheric phase and deformation phase signals in the unwrapped phase. In this paper, the phase ramps were estimated with a bilinear model (Shi et al. 2016). The DEM residual phase was estimated by the linear relationship

between topographic error and baseline. Temporally high-pass and spatially low-pass filters were employed to remove the atmospheric phase. Once sources of errors are mitigated, the subsidence rate and time-series displacement history were retrieved by a least-squares adjustment. We converted the LOS displacement to the vertical direction by means of dividing by the cosine of the incidence angle as Zhou et al. (2017) did in previous study.

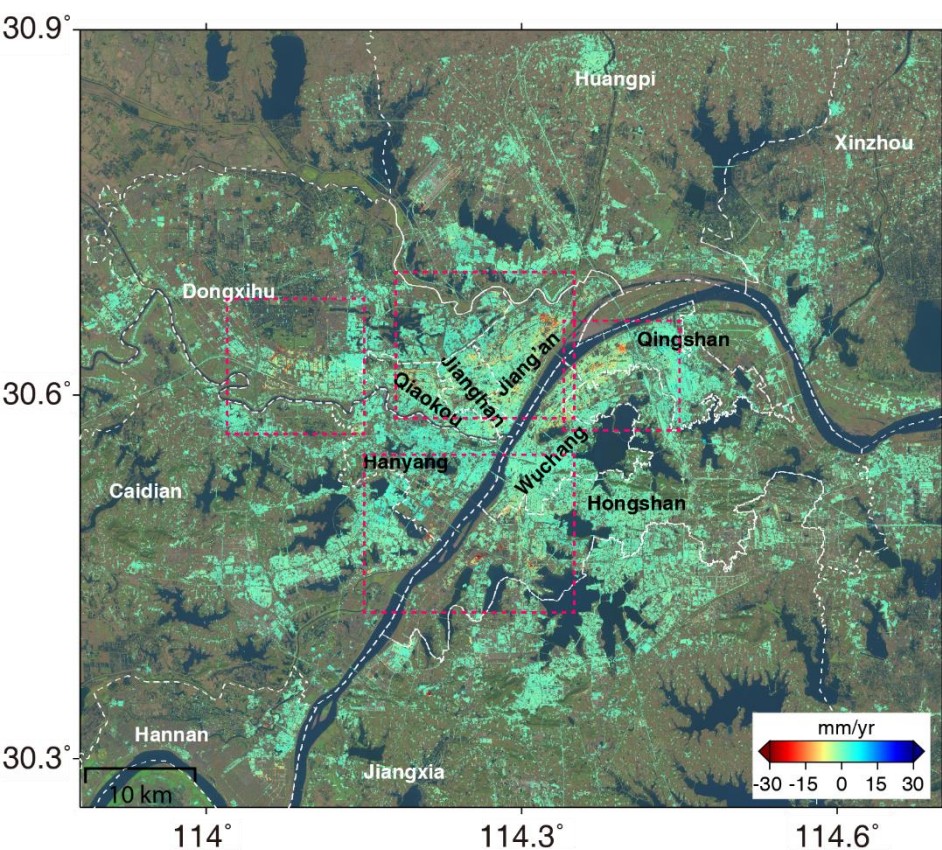

**Figure 3: Subsidence rate over Wuhan metropolitan area between 2015 and 2019. The background is Sentinel-2 image acquired at 6th, December 2019.**



## 4 Results

### 4.1 Mean subsidence velocity map

Figure 3 shows the subsidence rate derived from the Sentinel-1 SAR data. Given the dense man-made objects with stable backscattering signals over long time periods over urban scenarios, a total of 8,628,652 coherent pixels are selected. There are many factors (e.g. city constructions, karst landforms and soft soils) that cause ground subsidence in Wuhan. We can notice that the positions of subsidence in Wuhan metropolitan area are mainly distributed within EGSs that are composed of compressible soft soils in the first and second terrace shown in Fig. 1(b). We find the new localized subsidence centers have

emerged and old subsidence centers identified by previous studies have stabilized with decreased displacement rate. For example, the uplift areas in the right bank of Yangtze River and the subsidence center at Jianghan and Qiaokou district caused by construction of metro lines 1 and 2 during 2009-2010 (Bai et al. 2016) have almost stabilized in this study. Localized subsidence centers in Jiang'an, Qingshan, Hongshan, Wuchang, Hanyang and Dongxihu are identified which mostly cause by the intense anthropogenic activities, such as construction of metro lines and new buildings (Bai et al. 2019).

### 4.2 Comparision between InSAR and leveling measurments

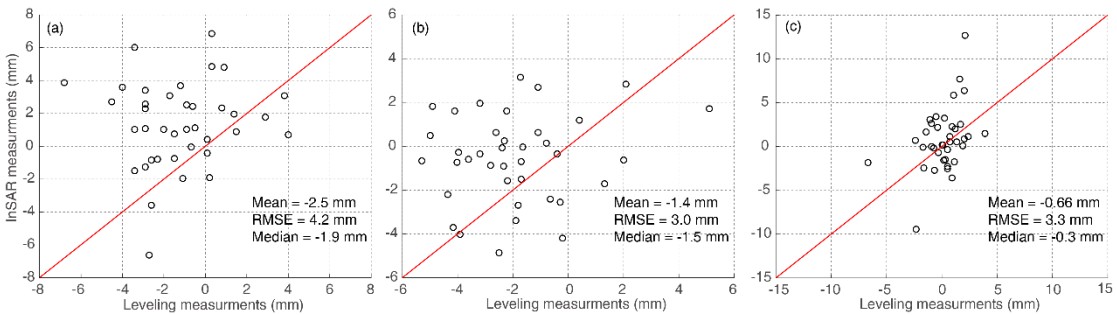

**Figure 4: InSAR measurements versus levelling measurements. (a) InSAR (20160827~20170309) vs levelling (20160910~20170310) (b) InSAR (20170309~ 20171009) vs levelling (20170309 ~ 20171010) (c) InSAR (20171009 ~ 20180513) vs levelling (20171010~**
**20180510)**

To quantify the results, we further compared InSAR with levelling measurements. We divided the levelling data into 3 time periods and compared them with InSAR data measured at closest dates as shown in Fig.4. The mean, root mean square error (RMSE) and median of the difference between InSAR and levelling measurements indicated our InSAR results reached millimetre accuracy. The statistical metrics in Fig. 4(a) which are slightly larger than these in Figs. 4(b) and (c) might be

caused by longer time coverage of InSAR than levelling.





## 4.3 Qingling-Jiangdi area

Karst collapse is listed as one of the most serious geological disasters in Wuhan (Wuhan Bureau of Natural Resources and Planning 2018). With the intensive anthropogenic activities in recent years, the occurrence of karst collapses or sinkholes in Wuhan has dramatically increased (Zheng et al. 2019). Fig. 5(a) gives the subsidence rate over Qingling-Jiangdi area
including Qingling town in Hongshan district, Jiangdi town in Hangyang district and Baishazhou and Zhangjiawan in Wuhcan district. In contrast with Fig. 5(b), 23 of the aforementioned 38 historical karst collapses occur in this area. The subsidence is identified at all the EGSs in the first terrace and the second terrace lacustrine EGS. The deformation of QL1 was correlated with karst subsidence (Bai et al. 2016). Anthropogenic activities are the main cause of subsiding in Qingling-Jiangdi area, e.g. QL2, QL3 and QL4. Fig. 5(c) and (d) are the optical images covering point QL3 obtained at 16th August,
2013 and 9th December, 2017 respectively. Constructions of metro line 6 depot (QL3) induced serious deformation with subsidence velocity over -30 mm/yr. The accumulative deformations of QL2, QL3 and QL4 reached over 160 mm from April 2015 to September 2019.

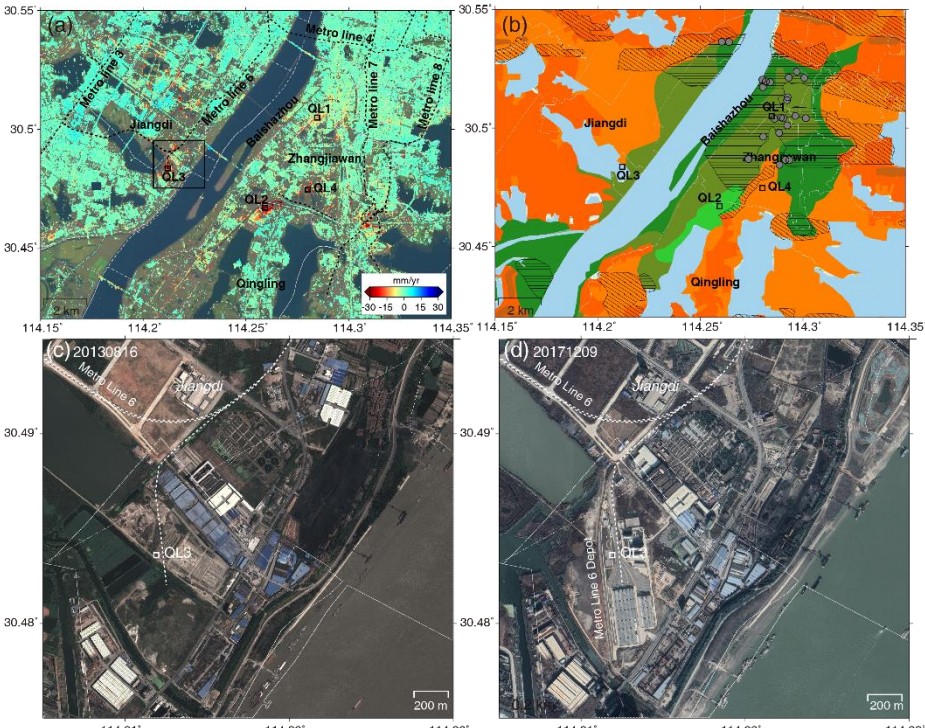

**Figure 5: (a) Subsidence velocity of Jiangdi–Qingling area during 2015 – 2019. The rectangle represent the location of (c) and (d).**
**(b) is the corresponding EGS map. The legend is the same as Fig. 1(b) while the grey circles are the historical karst collapses. The distribution of metro lines are freely available from ©Open Street Map contributors 2020. Distributed under a Creative Commons BY-SA License. (c) and (d) are ©Google Earth™ images covering point QL3 acquired from August 2013 and December 2019.**





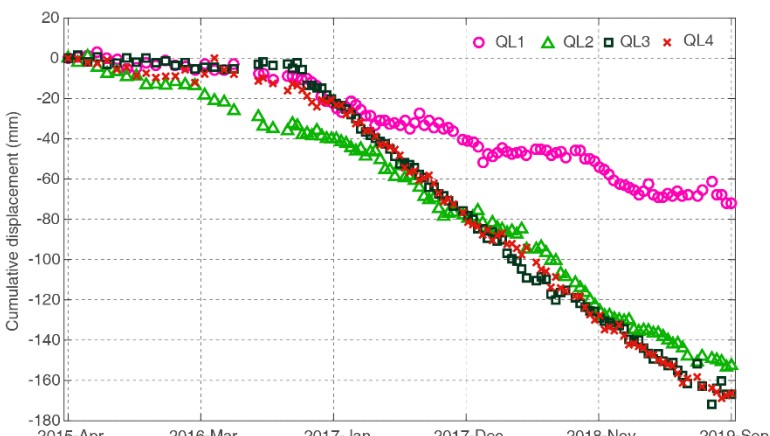

**Figure 6: Time series displacement of QL1~4 marked in Fig. 5.**

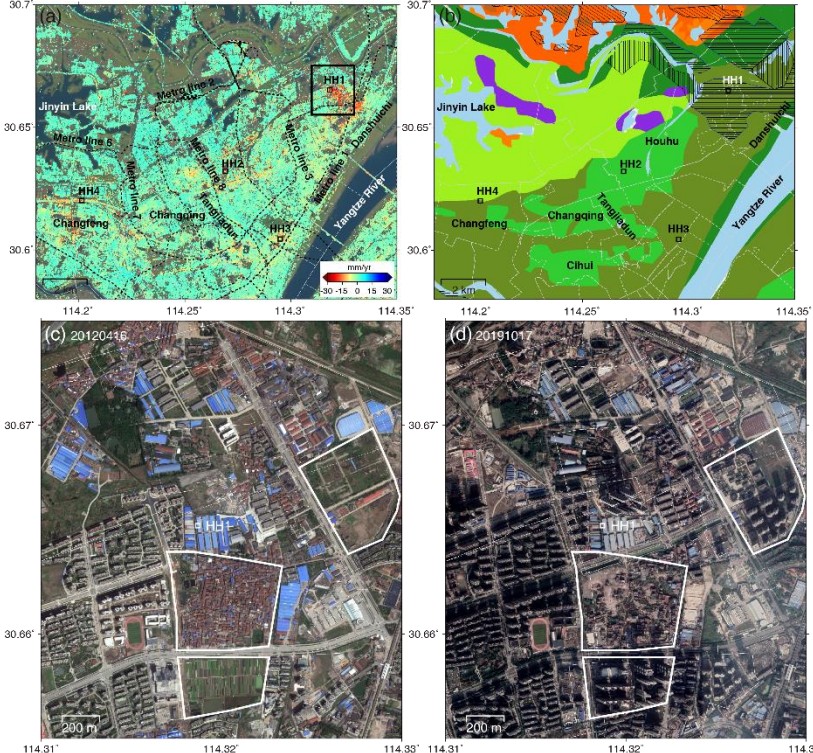

**Figure 7: (a) Subsidence velocity of Houhu area during 2015 – 2019. The rectangle represent the location of (c) and (d). (b) is the corresponding EGS map. The legend is the same as Fig. 1(b). (c) and (d) are ©Google Earth™ images covering point HH1 acquired from April 2012 and October 2019. The white rectangles marked the newly build-up areas.**




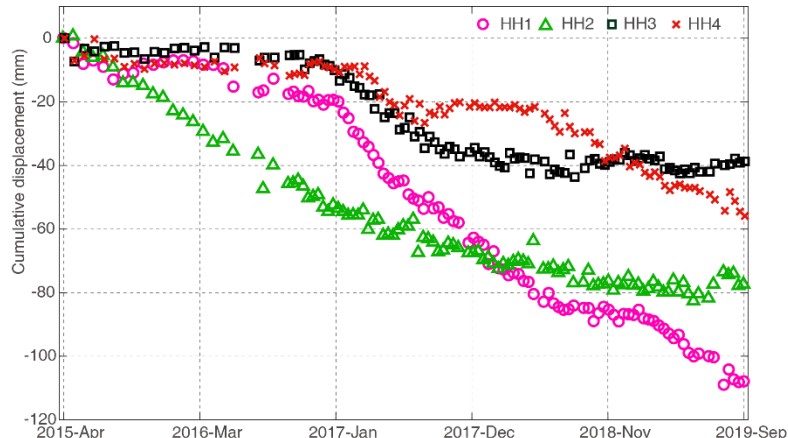

**Figure 8: Time series subsidence of HH1~4 marked in Fig. 7(a).**

## 4.4 Houhu area

Houhu area was originally lake or lake beach with underlying layers composed of highly compactable muck or muck soil with thickness ranging from 10~30 m (Sun et al. 2019). Due to activities such as ponds filling and construction of embankments, the surface was covered by soft artificial fill. Serious subsidence has occurred in this area due to the urban development in recent years (Wuhan Bureau of Natural Resources and Planning 2018). The subsidence of Houhu area located in Jiang'an district is very subtle during 2009-2010 with subsidence rate of -5~5 mm/yr (Bai et al. 2016) while a maximum subsidence rate with 86 mm/yr cantered at longitude 114.30 ° and latitude 30.6 ° was detected in 2013-2015 due to the construction of metro line 3 (Bai et al. 2019). However, the current velocity decreased to ~ 10 mm/yr during 2015- 2019 indicated by HH3 and a new subsidence center where HH1 located with displacement velocity of ~30 mm/yr was identified in our study as shown in Fig. 7(a). The localized subsidence centres shift with the urbanization progress. Fig. 7(c) and (d) shows the ©Google Earth™ images acquired at April 2012 and October 2019. The white rectangles indicated the newly built-up areas with maximum subsidence velocity of 20~30 mm/yr. The accumulative subsidence of HH1~HH4 are given in Fig. 8. The subsidence trends are all nonlinear which might closely correlated with the construction activities in the surrounding areas. The most serious accumulative subsidence occurred at HH1 which exceeded 100 mm during April 2015 to 2019 September.

## 4.5 Qingshan area

The subsidence rate and corresponding EGS map over Qingshan area are shown in Figs. 9(a) and (b). It is clear that the subsidence is mainly distributed in the EGSs covered by muck soil or soft clay. The subsidence center located in QS1 agrees with previous study that used RADARSAT-2 images from 2015 to 2018 (Zhang et al. 2019). The subsidence center (QS2) detected in our study is also identified from the results with TerraSAR-X datasets during 2013 -2015 (Bai et al. 2019). Fig.
9(c) and (d) show the ©Google Earth™ images of QS2 acquired at July 2014 and October 2019 respectively, in which the white polygons highlight the newly constructed areas. The construction activities in the surrounding areas of QS2 might

induced the subsidence which also the same for QS1 in Fig. S1(a) and (b). The accumulative subsidence in QS1 and QS2 presents significant nonlinear subsidence trends in Fig. 10. The subsidence rate might be affected by the construction intensities. Meanwhile, we also notice that burial carbonatite EGSs can be found over Qingshan area, which are mainly composed of dolomitic limestone and argillaceous limestone with high content of magnesium carbonate and argillaceous limestone, and difficult to dissolve (Xu, 2016). The subsidence rates are around 5 mm/yr. Meanwhile, the susceptibility of

karst collapse is considered to be low (Wuhan Bureau of Natural Resources and Planning 2018).

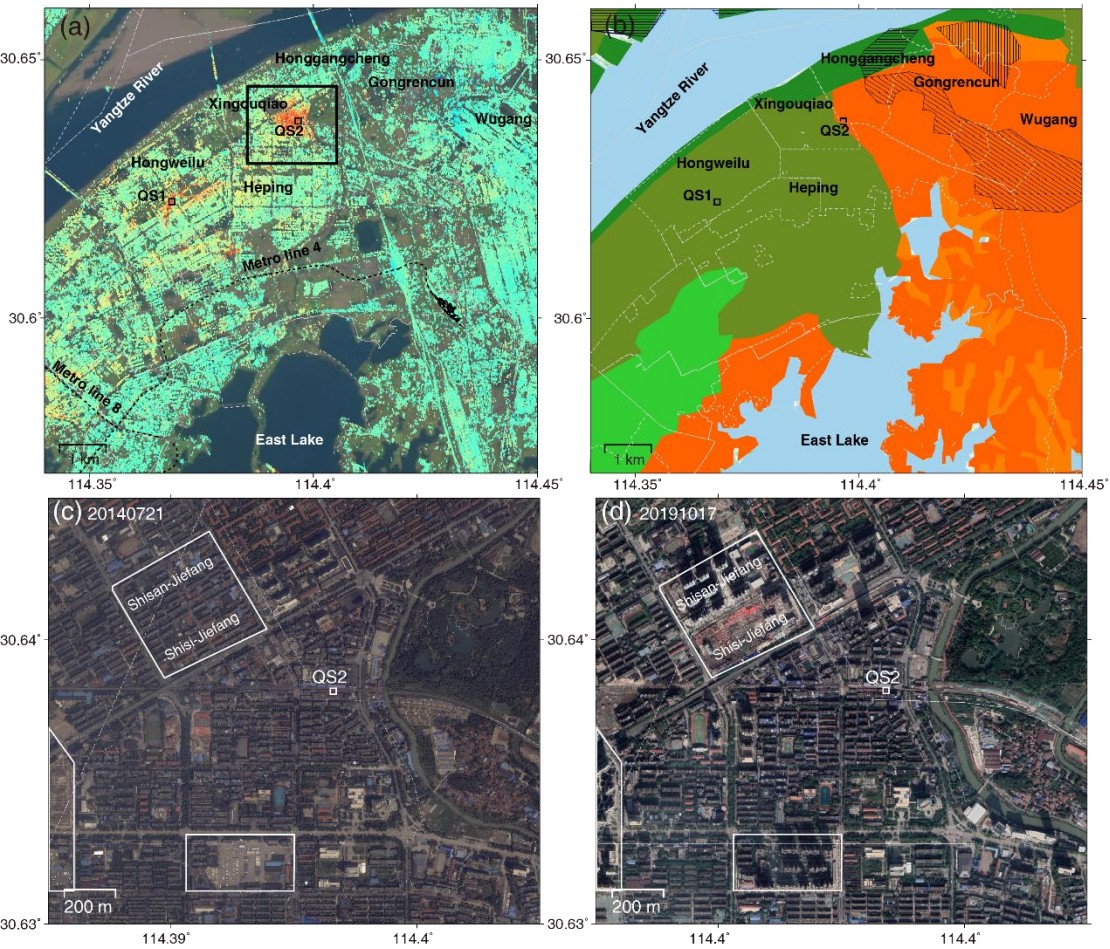

**Figure 9: (a) Subsidence velocity of Qingshan area. The rectangle represent the location of (c) and (d). (b) is the corresponding EGS map. The legend is the same as Fig. 1(b). (c) and (d) are ©Google Earth™ images covering point QS2 acquired at July 2014 and October 2019. The white polygons show the newly build-up areas.**





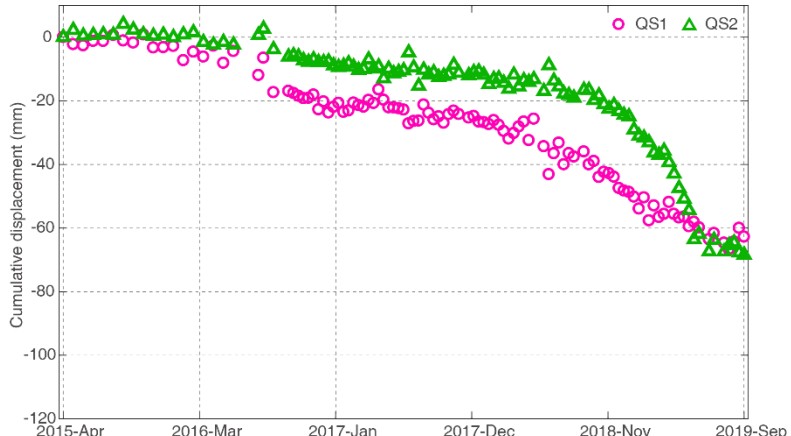


**Figure 10: Time series subsidence of QS1 and QS2 marked in Fig. 9(a).**

### 4.6 Dongxihu area

The subsidence rate over Dongxihu area is shown in Fig. 11, in which displacements are mainly distributed over the first terrace lacustrine EGS and first terrace alluvial EGS with flat terrain. The maximum displacement is up to ~30 mm/yr. The

subsidence of the carbonatite EGSs in this area was less than 10 mm/yr. The Fig. 11(c) and (d) show the ©Google Earth™ images covering D1 and D2 acquired at July 2013 and May 2019. We roughly labelled the ground changes over this period with white polygons. We can see that many land cover conversions occur in the surrounding area of D1 and D2. The ongoing constructions, together with the artificial loading are responsible for subsidence over this region. The accumulative subsidence of D1 and D2 show nonlinear displacement behaviours and reach approximately 80 mm between 2015 and 2019

in Fig. 12.




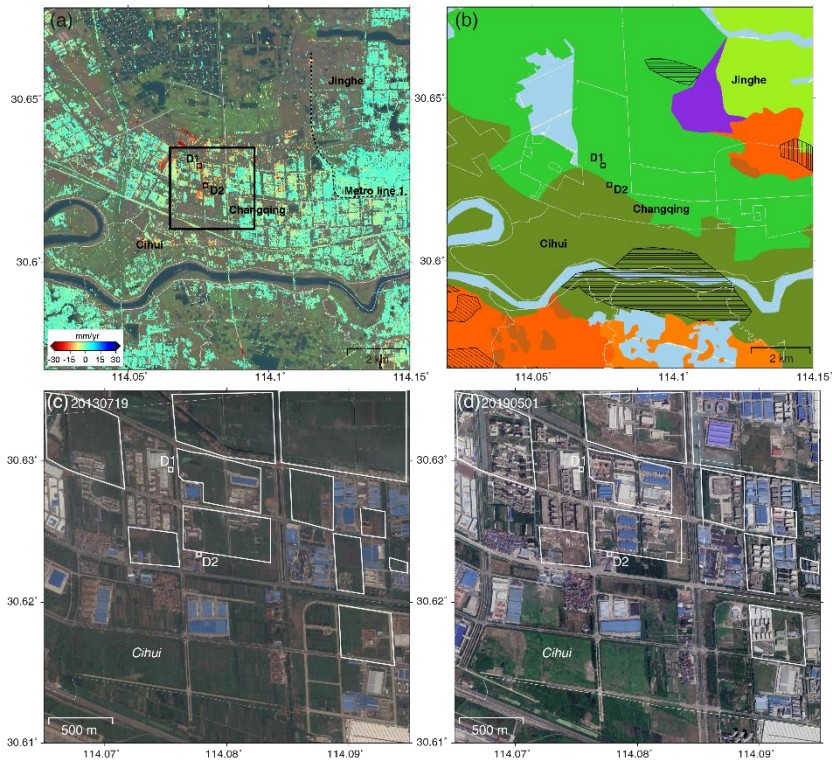

**Figure 11: (a) Subsidence rate of Dongxihu area during 2015 – 2019. The rectangle represent the location of (c) and (d). (b) is the corresponding EGS map with legends same as Fig. 1(b). (c) and (d) are ©Google Earth™ images of points D1 and D2 acquired at July 2013 and May 2019. The white polygons mark the changes over this period.**


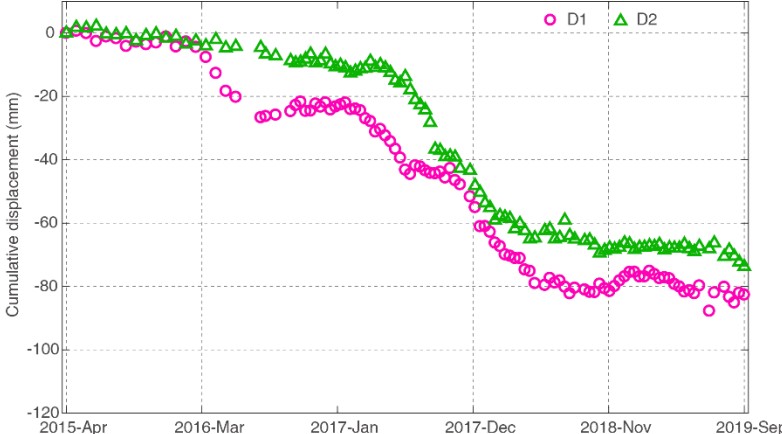

**Figure 12: Time series displacements of D1 and D2 marked in Fig. 11.**





## 5 Discussions

### 5.1. Groundwater pumping induced subsidence

Uneven settlements caused by excessive pumping of groundwater are common phenomenon in rapid developing cities (Chaussard et al. 2013, Wuhan Bureau of Natural Resources and Planning 2018). Thus, the groundwater extractions are restricted in Wuhan since 2013 (Chen, 2016). However, dewatering process is generally carried out during construction of deep foundation pits (Li et al. 2013, Cui et al. 2018). As a result, groundwater level might gradually decline in the surrounding areas and subsidence occurs in the meantime. During 2014, there were approximately 10,473 building sites and

539 municipal infrastructure construction site in Wuhan (Xu, 2016). As shown in Figs. 5, 7, 9 and 11, subsidence caused by construction activities is widely distributed in Wuhan. Taking QS1 as example, QS1 is about 150 m away from the Baoye Centre which was constructed since November 2015 in Fig.S1(a). Signs of construction was observed from the optical image acquired at February 2016 in Fig. S1(b).Accelerations from the time series subsidence was also observed in Fig. 10. Similar behaviours can be observed also from QS2, which is about 500 m away from the Shisan-Jiefang and Shisi-Jiefang.

Interpretation from the optical image acquired at May 2017 in Fig. S1(c), the housing demolition is almost finished. Constructions started since May 2018 and buildings are identified at Shisan-Jiefang from the optical images acquired at October 2018 Fig. S1(d). Accelerations can be clearly observed by InSAR measurements in Fig. 10 during the deep foundation pit dewatering. Meanwhile, the first terrace and second terrace EGSs are composed of typical binary structural stratums (Table S1) which are soft clay or muck soil in the upper part and sandy soil or sandy gravel in the lower part. The

groundwater level is highly correlated with the river levels of Yangtze River or the Han River, especially in the first terrace (Li et al. 2013, Chen 2016). Water level correlated displacement was also observed in the first terrace in previous studies using TerraSAR-X data (Bai et al. 2016).

### 5.2. Relationship between Karst subsidence and river water level/rainfall

Groundwater level variations may create negative pressure in fissures near bedrock surface of the karst caves (Wang et al.

2020). In general, the groundwater in karst area can be divided into the perched water, pore confined water, fracture karst water and fracture water from the top to bottom layers (Wang et al. 2020). The perched water and pore confined water are directly connected with the river level (Tu et al. 2019). In statistics, directly pumping of groundwater induced karst collapse did not occurred since 2001 due to the strict control by the government (Xu, 2016). However, anthropogenic engineering activities, e.g. foundation engineering or drilling, cause the occurrences of karst collapse frequently (Zheng et al. 2019). Also,

the karst collapses are closely correlated with the Yangtze River or the Han River water level fluctuation (Chen, 2016). QL1 in the Qingling-Jiangdi area and HH1 in the Houhu area were located in carbonatite EGSs in this study. To analyse the impact of water level or rainfall, we removed the linear displacement components from both points. The nonlinear part of the time series subsidence together with the rainfall and water level were depicted in Fig. 13. It can be seen from Fig. 13(b) that the river level are correlated with the rainfall to some extent. We can find that the nonlinear subsidence of QL1 is highly


correlated with the variation of river level. Acceleration occurs after the water level declines. Moreover, it is worth noting that there are lags between subsidence and river level fluctuation which is caused by the lags between river level and groundwater level (Chen 2016). On the contrary, the interaction between river level changes and subsidence is not remarkable at point HH1. We found that the dissolution degree of bedrock of the karst caves at Qingling-Jiangdi area is high while it is low for Houhu and Qingshan area (Xu 2016).The subsidence of HH1 might be dominant by construction activities.

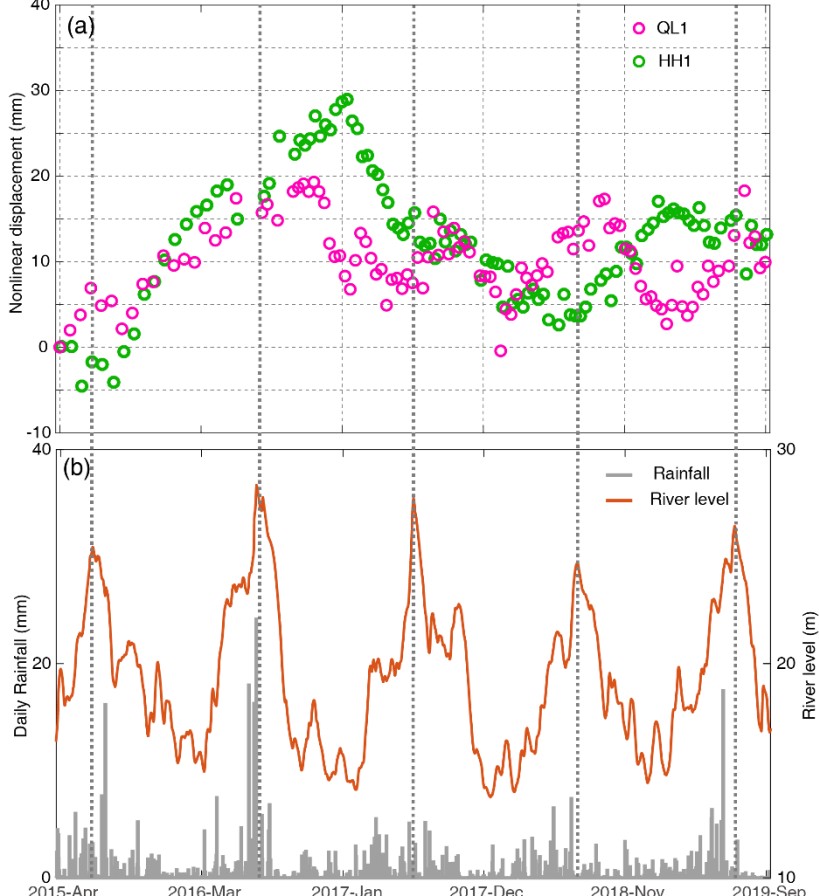


**Figure 13: (a) Nonlinear subsidence of HH1 and QL1, (b) Water level of Yangztze River and rainfall.**

## 6 Conclusions

We obtained the subsidence rate map over Wuhan region from Sentinel-1 imageries acquired from 2015 to 2019. Our results were validated with levelling measurements with accuracy better than 5 mm, indicating that InSAR is an effective tool in

monitoring ground subsidence with dense measurement points in urban areas. Our study revealed that the overall subsidence trends agree well with the distribution of EGSs covered by soft soils. The rapid urban development is the dominant impact factor of subsidence. Dewatering process of deep foundation pits and the corresponding consolidation lead to serious



subsidence, e.g. Qingling-Jiangdi area in Fig. 5, Houhu area in Fig.7 and Qingshan area in Fig.9. The subsidence centres shift with the intensity of urban constructions. Furthermore, we found that Qingling-Jinagdi area located in the first terrace is suffering from karst subsidence with velocity from 20~30 mm/yr, which brings great threats to peoples' daily lives. Comparison with precipitation and water level data, the nonlinear characteristic of karst subsidence is highly correlated with the water level fluctuations with lags. Nowadays, the advent of the European Space Agency's Copernicus program and the upcoming NASA/ISRO SAR mission provides unprecedented opportunities for continuous radar mapping of the Earth with enhanced revisit frequency, which help understand the underlying driving factors and detect anomalies in subsidence under the background of urban sprawl.

**Data availability.**

The Copernicus Sentinel-1 data were provided by European Space Agency (ESA) through the Alaska Satellite Facility (ASF). The precipitation data are provided by the China Meteorological Data Service Centre (http://data.cma.cn). The water level data of Yangtze River is collected from Hefei flood and drought information network (http://sq.hfswj.net:8000/Default.aspx)

**Author contributions.**

X.S., S.Z., M.J. and Y.P. conceived and designed the experiments; X.S. performed the experiments; X.S., T.Q., J.X., and C.Y. analysed the results; X.S. wrote original manuscript with the input from all co-authors.

**Competing interests.**

The authors declare that they have no conflict of interest.

**Acknowledgements.**

**Financial support.**

This work was financially supported by the National Natural Science Foundation of China (Grant No. 41702376) and the Key Research Program of Department of Education of Anhui Province, China (KJ2018A0503).



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
