# Peer review of "Spatial and temporal subsidence characteristics in Wuhan city (China) during 2015-2019 inferred from Sentinel-1 SAR Interferometry"

_Natural Hazards and Earth System Sciences, 2021_

## Author Comment (AC1)

RE: NHESS-2021-35

**We thank Deodato Tapete and two anonymous reviewers for their invaluable comments that have helped improve our manuscript. Our responses to the comments are highlighted in bold. Please refer to the tracked-change version for the line numbers addressed in this letter.**

RC1 (Anonymous Referee #1)

This paper evaluated the subsidence of Wuhan during 2015-2019 with Sentinel-1 InSAR dataset. They find the distribution of deformed areas are spatially correlated with engineering geological regions and rapid urbanization. Moreover, they discovered the time series displacements of karst areas are affect by the Yangtze water level variations. This research fits the scope of *NHESS* and I suggest a minor revision. My detailed comments are listed as follows:

1.In Section 4.6,©Google Earth™ images are acquired at July 2013 and time-series analysis starts from 2015.4,Whether the construction date can be explained in detail in order to better explain the accelerated deformation(As described in line 252-257).

**Yes, we updated Figure 12 and corresponding text in section 4.6 (Line 251-255) to better explain the accelerations. Optical images acquired in August 2016 and December 2017 are shown in Figure 12(c) and (d). The also updated Fig. 8.**

2. Line 261, the authors proposed water level correlated displacement might exist in the first terrace. Can you show us some examples?

**Previous study by Li et al. (2013) and Chen (2016) indicated the groundwater level in the first terrace correlate with river level. Han et al. (2020) and (Bai et al. 2016) also identified water level correlated displacement in the first terrace covered by soft soils. In our opinion, the water level related displacement should exist along banks of rivers.**

**PS pixels located on natural ground can be selected to analyze the interaction between subsidence and river water level. However, the bank area was flooded in July 2016 caused by concentrated rainfall. As a results, very sparse pixels are detected on the natural ground as we can see from in Figure 8(a) and 10(a).**

**The pixels we selected on manmade structures as shown in Fig. R1 at the bank of Yangtze River shows obvious seasonal signal which might correlated with river level. However, we cannot exclude the thermal impact caused by temperatures which is very common in manmade structures. Therefore, the relationship between displacement and water level in the first terrace were not given in the manuscript.**

[Figure]

**Fig.R1 (a) Cumulative subsidence of selected points P1 (lon=114.2749, lat = 30.5100) and P2 (lon0 = 114.2369; lat0 = 30.4634), (b) Water level of Yangtze River and rainfall.**

3. The authors should carefully check the type errors. The legend and scale in subsidence rate map should be consistent (eg. section 4.5, 4.6).

**We check our manuscript carefully and correct all the type errors and updated figures.**

4. Line 279, "The subsidence of HH1 might be dominant by construction activities." After 2017-Dec, the interaction between river level changes and subsidence is not so remarkable at point HH1, can you describe the construction activities details or activities which were different from QL1?

**In our opinion, HH1 is affected by human activities and QL1 is correlated with rainfall or river water level. We rearranged section 5.2 (Line 300-312).**

**The subsidence of HH1 is dominated by continuous construction activities which can be inferred from Fig. 7(c) and (d) and Fig. S4. Many land conversions occurred during 2015 and 2019 in Fig. 7(c) and (d). The subsidence of HH1 was caused mostly by the deep foundation pit dewatering. Construction activities were observed at the area marked by the red rectangle located extremely near HH1 during February 2016 and December 2017. Although we don't know the exact data of construction activities, accelerations was observed at HH1 after January 2017.**

The construction intensities at QL1 are extremely low than that of HH1 during 2015-2019. At the meantime, many of karst collapse caused by natural factors, such as rainfall and water level, were observed in Qingling area. The displacement of HH1 was originally very small before 2016. The trigger factor of the accelerations might be the extremely rainfall. Since then, the displacement of HH1 presented clearly seasonal patterns.

---

## Author Comment (AC2)

**RE: NHESS-2021-35**

**We thank Deodato Tapete and two anonymous reviewers for their invaluable comments that have helped improve our manuscript. Our responses to the comments are highlighted in bold. Please refer to the tracked-change version for the line numbers addressed in this letter.**

CC1 (Deodato Tapete)

The present manuscript focuses on the spatial and temporal distribution of land subsidence hotspots across the expanding and developing urban footprint of Wuhan in China.

At the moment there is a growing InSAR literature investigating land subsidence and karst collapse hazard in Wuhan. Therefore, the authors should contextualise their results and compare with published InSAR results achieved by processing either (nearly) the same Sentinel-1 dataset used in this paper or other SAR datasets.

It is with regard to this important aspect that my comment is made.

The authors seem not to have accounted for the following study:

Han, Y.; Zou, J.; Lu, Z.; Qu, F.; Kang, Y.; Li, J. Ground Deformation of Wuhan, China, Revealed by Multi-Temporal InSAR Analysis. Remote Sens. 2020, 12, 3788. https://doi.org/10.3390/rs12223788

Han et al. (2020) have processed and analysed basically the same Sentinel-1 dataset i.e. April 2015 to June 2019, with SBAS-InSAR technique. So, there is a straightforward opportunity for the authors of the present manuscript to make a comparative discussion of their results with those published by Han et al. (2020).

Another very recent paper that the authors should also consider is:

Jiang, H.; Balz, T.; Cigna, F.; Tapete, D. Land Subsidence in Wuhan Revealed Using a Non-Linear PSInSAR Approach with Long Time Series of COSMO-SkyMed SAR Data. Remote Sens. 2021, 13, 1256. https://doi.org/10.3390/rs13071256

In this paper, my collaborators and I have processed and analysed the longest time series of COSMO-SkyMed data that has been published so far over the city of Wuhan.

Because our paper and the present manuscript share the common interest on correlating the observed land subsidence with soft soil consolidation, it would be interesting if the authors would enrich the discussion of their results vs. those published

in our paper.

**Thanks for pointing this out. Both papers are important studies in the literature. Han et al. (2020) used Envisat ASAR (2008-2010), ALOS PALSAR (2007-2010) and Sentinel-1 (2015-2019) data to study the spatial displacement characters of Wuhan. They identified the displacement trend significantly decreased in 2017. Jiang et al. (2021) used long-term and consistent high resolution CSK dataset acquired from 2012 and 2019 to study the displacement of Wuhan using the non-linear PS-InSAR approach. They found accelerations of ground displacement correlated with construction activities. They also identified the 2016 heavy rainfall events caused accelerations. Both research found the displacement correlated with soft soil consolidation. Our results agreed with both of the studies. The findings of these two papers are properly cited in the introduction and results section of our manuscript.**

Further line-by-line comments are appended here below:

- Lines 34 - 44: these sentences are very common knowledge for the journal readership and can be removed, alongside the cited references. This should help the authors to shorten the manuscript and save space for the discussion later on.
**We remove these sentences and references in the manuscript accordingly.**

- Lines 52-54: with regard to the mention of COSMO-SkyMed, the whole archive of COSMO-SkyMed 2012-2019 has been analysed and very recently published by Jiang et al. (2021) - see comment above. This should be acknowledged to keep the state-of-the-art section updated with the very recent literature
**We add the recent works by Jiang et al. (2021) and Han et al. (2020) in the state-of-the-art section (Line 56 and 59-60).**

- Figure 1: karst collapses and levelling points are barely visible. The authors should consider the addition of a zoomed view.
**We updated Figure 1 by adding a zoomed view in Figure 1(c).**

- Line 74: The authors should specify what "The Rise of Central China" is.
**"The Rise of Central China" is a policy to accelerate the development of central China including Shanxi, Henan, Anhui, Hubei, Hunan, and Jiangxi. We add this information in Line 80.**

- Lines 112-116: these sentences are very common knowledge for the journal readership and can be removed, alongside the cited references. This should help the authors to shorten the manuscript and save space for the discussion later on.

**We remove these sentences and references in the manuscript accordingly.**

- Line 124: why did the authors choose 500 m as the upper limit of bperp, given that Sentinel-1 ref. tube deviation is +/- 100 m (https://sentinels.copernicus.eu/web/sentinel/missions/sentinel-1/satellite-description/orbit)?

**Thanks for pointing this out. With the good orbit control ability of Sentinel-1, a 500 m limit of perpendicular baseline is meaning less in this study. Thus, we remove this statement. Only temporal baseline less than 60 days are used (Line 133).**

- Line 125: The section lacks of information about the software that has been used to process Sentinel-1 data or, instead, if a proprietary code has been used.

**We processed the Sentinel-1 interferometric data using software published in the journal Computer & Geosciences and list as an reference, which is illustrated in Line 128-129.**

**Reference:**
**Yu, Y., Balz, T., Luo, H., Liao, M., and Zhang, L.: GPU accelerated interferometric SAR processing for Sentinel-1 TOPS data, Computers & Geosciences, 129, 12-25.**

- Line 148: this spatial intersection should be better displayed by combining the InSAR subsidence rates and geological datasets in the same figure.

**The readability might be reduced if we superpose the displacement rates on colored geological map. Thus, we superpose the boundary of the first terrace EGZ and second terrace EGZ on the displacement map to roughly illustrate the correlation between EGZ and displacement and updated Figure 4.**

- Section 4.4. Houhu area: how do the present results and time series compare with those published in Han et al. (2020) at equal SAR data processed?

How with Jiang et al. (2021) who processed X-band high resolution data with non-linear PSInSAR technique?

**Our study agreed with results from Han et al. (2020) and Jiang et al. (2021). The long term displacement from Jiang et al. (2021) indicated widely distributed**

subsidence occurred during 2012-2019 which might corresponds to different nonlinear subsidence phase. At the meantime, the localized subsidence center identified in our study coincide with Han et al. (2020) with same order of displacement rate. The results in our study enable us to identify short-term (2015-2019) localized subsidence centers. (Line 211-213 and 218-219)

- Section 5.2, Relationship between karst subsidence and river water level/rainfall: how do the present results compare with those published in Han et al. (2020)?

Han et al. (2020) found that the changes of land subsidence near the bank of the Yangtze River are generally consistent with the variations in the river water level over most of the monitoring period. However, they also noted a time delay with respect to the time of water level changes, suggesting the complexity of and variation in the hydrogeological condition along the Yangtze river in Wuhan. What is the authors' opinion in this regard based on their data?

**The groundwater level in the first terrace might correlate with river level (Li et al. 2013, Chen 2016). Han et al. (2020) and (Bai et al. 2016) identified water level correlated displacement in the first terrace covered by soft soils. In our opinion, the water level related displacement should exist along bank of rivers.**

[Figure]

Figure C1. Typical land cover along the bank of Yangtze River from @Google earth™ image.

**In our opinion, PS pixels located on natural ground should be selected to analyze the interaction between subsidence and river water level. The bank area was flooded in July 2016 as shown in the Fig. C1. Although a SBAS workflow which can make use of distributed scatterers was employed in our study, the pixels we selected in this study were mainly manmade structures. Very sparse pixels are detected on the natural ground as we can see from in Figure 8(a) and**

**10(a).**

**The pixels we selected on manmade stuctures at the bank of Yangtze River shows obvious seasonal signal as Han et al. (2020) did in their study which might correlated with river level. However, we cannot exclude the thermal impact caused by temperatures which is very common in manmade structures. Therefore, we skipped this part in our manuscript.**

[Figure]

**Fig.C2 (a) Cumulative subsidence of selected points P1 (lon=114.2749, lat = 30.5100) and P2 (lon0 = 114.2369; lat0 = 30.4634), (b) Water level of Yangtze River and rainfall.**

**The displacement of QL1 in karst areas did not observe the thermal impact and construction intensities are low as we can infer from Fig. 14. As pointed out by reviewer 2, the displacement of QL1 is more correlated with rainfall rather than river level. We updated section 5.2.**

- Lines 292-295: please revise this last sentence in the context of the future direction of the present research.

**Agreed and revised.**

---

## Author Comment (AC3)

RE: NHESS-2021-35

**We thank Deodato Tapete and two anonymous reviewers for their invaluable comments that have helped improve our manuscript. Our responses to the comments are highlighted in bold. Please refer to the tracked-change version for the line numbers addressed in this letter.**

RC2 (Anonymous Referee #2)

Land subsidence is one of most common geohazards. It is significant for monitoring the characteristics for city. This paper uses the time series interferometry technology to obtain the spatial and temporal subsidence characteristics in Wuhan city (China). The results indicate that the overall subsidence over Wuhan region is significantly correlated with the distribution of engineering geological regions. The results sound good. I recommend a minor revision. The detailed comments are as follows.

(1) In section 4.2, the InSAR measurements are compared with leveling measurements. Fig. 4a and 4b indicate a bit lower correlation value. Please make a detailed analysis for the reasons. The authors can present some detailed InSAR results for some typical leveling points.

**Sorry for the confusion. We updated Figure 5(a) and (b). The leveling points are mainly distributed in the Wavy hillocky EGZ with low displacement rates. The agreement between InSAR and levelling is well as shown by the statistical metrics in Fig.5.**

(2) For the regions with larger deformation, the authors can add some field survey pictures (such as buildings with crack, road with cracks).

**We add two filed survey pictures in Houhu area and Qingshan area in Figure S1.**

(3) In section 2.2, the authors should present a flowchart for the used time series InSAR method.

**The flowchart is given in Figure 2.**

(4) In section 5.2, I think it is not necessary to compare the subsidence with river water level. According to the Fig. 13, I think the daily rainfall is more correlated with the nonlinear subsidence of points HH1 and QL1.

**Thanks for pointing this out. The impact factor of QL1 is rainfall. We rearranged section 5.2 (Line 300-314).**

**There are several impact factors of karst collapse including rainfall, water level variations and anthropogenic activities. Although HH1 located in karst distributed areas, the displacement of HH1 is more correlated with**

anthropogenic activities. Deep foundation pit dewatering induced groundwater variations induced the accelerations. As we can infer from Fig. 7(c) and (d) and Fig. S4, many land conversions occurred during 2015 and 2019.

QL1 was almost stable before summer of 2016. We find the trigger factor of QL1 might be the extreme rainfall in 2016 as reminded by the reviewer. The rainfall might played a more important role. At the meantime, we also notice the river water level is correlated with concentrated rainfall during rainy seasons in Wuhan. We cannot rule out the impact of water level. Thus, we kept the river level in our figure and revised the discussion part accordingly.